# Influence of Age on the Success of Neurorehabilitation

**Nicolas Broc** [1,*] and **Armin Schnider** [2]

1. Division of Neurorehabilitation, University Hospital of Geneva, 1205 Genève, Switzerland
2. Division of Neurorehabilitation, University Hospital and University of Geneva, 1205 Genève, Switzerland
* Correspondence: nicolas.broc@hcuge.ch; Tel.: +41-(0)22-55-37-885

**Abstract:** There is a general understanding that older adults suffering from a stroke have poorer outcomes and might benefit less from neurorehabilitation. This narrative review analyzes the conflicting evidence for the effect of aging on the success of neurorehabilitation after a stroke. While there is convincing evidence that functional outcomes are negatively impacted by age, functional gains made during rehabilitation are less clearly impacted, and the effect of age seems to be related to other factors such as prestroke independence and therapy intensity, as well as the population studied. There is no evidence that would justify withholding high-intensity neurorehabilitation on the sole basis of age.

**Keywords:** rehabilitation; ageing; outcome prediction; activities of daily living; stroke

## 1. Introduction

The aging of the general population is a challenge for health systems because of the increase in comorbidities, costs, and use of healthcare. Stroke neurorehabilitation is no exception, with age being an independent risk factor for both strokes and cardiovascular disease in general. There are stereotypical views as to which older patients benefit less from intensive neurorehabilitation and increasing pressures from health systems and insurance companies to rationalize care to favor only the most efficient health interventions. If advancing age is a negative factor for the outcome and efficacy of neurorehabilitation, this could serve as a rationale to limit access to inpatient neurorehabilitation to older patients.

The objective of this narrative, non-systematic review is to investigate the effect of chronological ageing on the success of neurorehabilitation after a stroke.

We conducted a literature search in PubMed using MeSH terms "(Stroke) AND (Neurological Rehabilitation) AND (Outcome Assessment, Health Care) AND (Aged, 80 and over)" in PubMed and then screening abstracts for relevance. The references of the two largest existing reviews from Jongbloed [1] et al. and Black-Schaffner [2] et al. were also screened for additional relevant studies.

## 2. Influence of the Studied Population

The context of the study and the selection of patients, which may vary in terms of age, functional impairment at baseline, and LOS, renders simple generalization of results difficult.

For example, a small case series conducted in Singapore [3], studying only patients older than 75 years old with a mean age of >80 years old, showed that 89% returned home and showed significant improvement in the activities of daily life (ADL) function. However, it is noted that "it is still culturally and socially unacceptable in Singapore for children to send their parents/relatives to nursing institutions" and that "(p)aid caregivers in the form of foreign maids are generally available and more affordable than nursing home costs". This certainly influences the high rates of return to home in this study. Also, the criteria for admission to the rehabilitation facility are listed as "ability to participate in therapy,

medical stability, and potential for functional gains", which is likely to have selected a patient population more likely to benefit from rehabilitation.

A study of stroke rehabilitation of patients aged >65 years old, in a Japanese acute neurology/neurosurgery ward [4] explored a relatively unselected population of patients, whose admission to a rehabilitation hospital was independent of their rehabilitation potential. By stratifying the study population in three age groups of 65–74, 75–84, and ≥85 years, they showed that the discharge functional impairment, measured with the functional independence measure (FIM), was reduced as a function of age, with a median FIM at discharge of only 44 for the oldest age group, compared to 103.5 and 74 for the 65–74- and 75–84-year-old groups, respectively. In parallel, rehabilitation efficacy was measured with an method coined Montebello rehabilitation factor score (MRFS), defined as $\frac{\text{discharge FIM} - \text{admission FIM}}{\text{maximum possible FIM} - \text{admission FIM}}$ to adjust for the ceiling effect of the measure. A drastic reduction of the MRFS was observed with age with 0.44, 0.21, and 0.08, respectively, in the three age groups. It is possible that the patients of this study included those severely affected older adults often not included in studies conducted in rehabilitation hospitals. Of note, the older patients were more often of female sex, premorbidly dependent, suffered from more severe motor paralysis, spasticity, sensory disturbance, and a range of motion restriction, trunk dysfunction, neglect, aphasia, and lower non-paretic limb function. The mini-mental state examination (MMSE) and the total, motor, and cognitive FIM are lower. The length of stay was longer, and the older group was less likely to return home.

The authors then performed multivariate regression analysis to assess the factors associated with discharge FIM and MRFS in the different age groups. There are shared and differing factors. For discharge FIM, while there were age-group specific factors (i.e., trunk dysfunction and neglect for the 65–74 age group), premorbid dependence, motor paralysis, and cognitive FIM were shared by the three groups. Rehabilitation efficacy depended on recurrent strokes and motor paralysis in the younger group, and on cognitive FIM and non-paretic limb function in the older age group.

## 3. Worse Functional *Outcomes*—Similar Functional *Improvement*

There is consistent evidence demonstrating a negative correlation between age with functional outcomes after a stroke. This has been extensively studied and reviewed since the eighties [1,2,5]. However, there is still ample debate as to the extent of this influence in quantitative terms, whether neurorehabilitation, nonetheless, influences this outcome, and about the mechanism of this effect.

We observed the three most common ways to study the effect of aging on the functional outcome after a stroke. The first one is by analyzing its relationship to discharge functional status or at different set time points, for example, after 3, 6, or 30 months. The second one was by investigating changes in functional status during the rehabilitation stay, often termed *rehabilitation efficacy*, and sometimes expressed as the proportion of possible improvement of the functional index that the patient achieved, to avoid the ceiling effect of the functional scales. Lastly, some authors looked at rehabilitation *efficiency*, defined by function points gained per day.

In the early review from Jongbloed [1], out of 14 studies reporting a negative effect of age, 13 do so by studying its effect on functional status at or after discharge from rehabilitation, whereas the four studies showing no effect of age studied the improvement in function. Only the study by Jimenez et al. [6] demonstrated a negative effect on the improvement of function. Lehman et Al. [7] found age to be negatively correlated with discharge function, but to have no association with *improvement* in function. It is noted there is no clear evidence whether age directly influences outcome or does so through associated diseases. There is great heterogeneity in the scales used to assess function, in the time points studied, and the sample sizes are generally small—some treating age as a continuous variable, while others as discrete age classes.

A later study by Black-Schaffner et al. [2] reviewed 13 studies between 1986 and 2004 and noted a shift towards the use of validated functional scales, mostly the FIM

(functional independence measure), an 18-item measurement tool with a 7-point ordinal scale (min. 0–max. 126) of established validity and interrater reliability, as well as the Barthel Index and its modified form (min. 0–max. 100). The sample size of the studies are larger, and the statistical analysis methods used more uniform. Again, there is no consensus about the influence of age *per se* but an agreement that functional level on admission to neurorehabilitation is an important predictor of discharge functional level and that admission functional level is lower in older adults.

The only two studies that assessed premorbid ADL function were by Colantonio et al. from 1996 [8] (premorbid Katz and Rosow scale, premorbid cognitive impairment) and by Mutai et al. from 2018 [4] (prestroke modified Rankin scale). None of the included studies assessed prestroke FIM.

In the study by Mutai et al., prestroke dependence (mRS 3–5) was associated with lower FIM at discharge from the hospital in all age groups and with rehabilitation efficacy, but only in the intermediate age group. The study by Colantonio showed that ADL function at 6 months post-stroke was associated with prestroke ADL functional impairment in the uni- and multivariate analysis. The risk of institutionalization after discharge was associated with prestroke cognitive impairment and impairment in the Rosow scale in uni- and multivariate analysis, but not prestroke ADL function using the Katz scale. There was no effect of age on function and institutionalization after stroke.

There are more studies looking into length of stay (LOS), discharge disposition, and rehabilitation *efficacy* and *efficiency*. Interestingly, Bagg et al. [9] demonstrated, prospectively, that age explains only 1% of the variability in outcome measured by total FIM or motor FIM at discharge, as opposed to functional independence at admission, which explains 15–66% of the variability. Furthermore, in this study, age had no effect on the change in FIM score during the hospital stay. It is noteworthy that, while most studies evaluate the effect of rehabilitation by studying outcomes after discharge or a certain period of inpatient rehabilitation, some studies evaluate time points after acute hospital discharge, where it is unclear what proportion of patients actually had inpatient rehabilitation.

## 4. Baseline Independence Is a Better Predictor of Outcome Than Age

The study by Black-Schaffner et al. [2] introduced another interesting idea, by stratifying patients into three groups according to their admission FIM (<40, 40–80, >80), thus being able to demonstrate a negative effect of increasing age on the total FIM change and the percentage of home discharge in all but the least functionally impaired group at baseline (FIM > 80). There is an interesting negative effect of increasing age on the LOS, only for the most functionally impaired (FIM < 40). Rehabilitation efficiency (the ratio of FIM change to LOS) shows a negative association with age in all but the least impaired group, which is particularly interesting when considering that this is despite the shorter LOS for the FIM < 40 group. The authors conclude that there is a subgroup of older adults that benefit equally from neurorehabilitation, suggesting that it is not age, but other factors that drive the apparent negative association with increasing age. Concerning length of stay, there is important heterogeneity in the literature, as it reflects many nonmedical factors like the health system, local practices, and philosophies. It is noteworthy that this study was conducted in an American long-term acute care rehabilitation hospital that requires a minimum stay of 25 days, as opposed to acute rehabilitation hospitals which have no LOS requirement.

## 5. Long-Term Retainment of Therapy Benefits

Most of the aforementioned studies followed patients for the length of their inpatient stay; a minority followed patients at fixed time points during or after their rehabilitation. A small study from Korea [10] studied patients up to 30 months after stroke: while the younger patients (20–69-year-old group) improved in terms of functional impairment for up to 6 months (measured with the mBI) and retained their improvement, the older group (70–89 years) improved only for 1 month and failed to retain their improvement between

6 and 30 months post-stroke. This study also showed the results for an ambulation scale (FAC), motor strength (measured with the Medical Research Council (MRC) Scale for Muscle Strength of the upper and lower extremity) and a scale of cognitive impairment (MMSE). There was a plateau in the older patient group between 6 and 30 months, it did not regress. Therefore, the secondary regression in functional impairment in the older group of this study cannot be explained by a secondary regression of gait, strength, or cognitive function.

## 6. The Importance of Therapy-Intensity

Few of the included studies specified *therapy intensity*, defined as "time spent in an active session" as an explanatory variable.

A large prospective database study by Knecht et al. [11] in a German Neurorehabilitation hospital, analyzed data from a 2294-patient cohort, by studying functional impairment measured with the Barthel index (BI) in three age groups of <65, 65–79, and >80 years old. Results referred to a fixed 4-week period to adjust for the varying length of stays. Therapy intensity in hours was extracted from the medical records. The authors found that the overall functional improvement (BI at discharge vs. admission) was the same for the three age groups. The amount of therapy predicted recovery and the relationship between the intensity of therapy and recovery was not influenced by age. In other words, the recovery gain per hour of therapy was the same in the older as in the younger group. Finally, they used item-wise logistic regressions to test for potential age-group differences in the recovery of each of the 10 functional domains assessed by the BI, to assess whether the odds of achieving an independent level of function for a given domain (i.e., achieving a maximum score on the corresponding BI item) was dependent on age, adjusting for domain-specific, and overall functional status at admission and administered therapy hours. There was no difference between the three age groups. The authors concluded by reflecting on the health economics point of view, estimating that "payback time for high-intensity neurorehabilitation for patients older than 80 years is less than half a year", taking into account data from a study [12] reporting that a "15 point difference in the BI ( . . . ) corresponds to a 30% reduction in stroke-related costs", and that "the average payback time for 59 days of inpatient neurorehabilitation (with an average improvement of 34 points on the BI) in terms of savings in post-rehabilitative nursing care is 21 weeks" [13].

None of the included studies mentioned *load* as a measure of "*intensity* (hours) × *Rate of Perceived Exertion*", and few studies do, in general. The most widely used measure of exertion is the Borg Rating of Perceived Exertion Scale, but there are also physiological measures like percentage of heart rate reserve or even more advanced physiological measures, as exemplified by a recent review from Mah et al. [14] on High Intensity Exercise on Lower Limb Function. Mostly, this approach equates *exertion* to cardiovascular effort and not necessarily to *difficulty* of a certain task or therapy, which is a multidimensional concept.

It would be interesting to know if this factor of *load* or therapy *difficulty* varies with increasing age and it can be postulated that the capacity for effort is more limited in the elderly, which might limit participation.

## 7. Health Disparities in Stroke Care for Older Adults

All authors conclude that neurorehabilitation should not be withheld based on age as a stand-alone factor, as the effect of other associated factors is much greater. Still, older adults often do not benefit from the same care as younger ones.

There is a great heterogeneity in rehabilitative care for older adults, especially when considering the relatively homogeneous acute stroke care since the implementation of stroke centers.

In a multicenter study in 10 European countries [15], 1,847 consecutive patients with a first stroke were stratified according to their age in a <75-year-old and >75-year-old group. Older patients "tended to be discharged earlier from hospitals and be institutionalized at higher rates compared with younger patients" and had less access to rehabilitation. A

database study in Canada [16], including 26,676 patients with ischemic stroke, stratified by age in five age-groups, showed that older patients had higher case-fatality, a longer length of stay, and returned less often to their original place of residence, while also being less admitted to the ICU. Notably, the older subgroups were more often treated at non-academic hospitals than the younger ones.

A study by Vluggen et al. [17] assessed the effect of a multidisciplinary geriatric rehabilitation program for older persons with stroke on the primary outcome measure of daily activity using the Frenchay Activity Index (FAI) scale. The authors found no significant improvement in the primary outcome measure. Nonetheless, it was found that patients participating in the program had a higher level of perceived autonomy of outdoor activities (using the Impact on Participation and Activity (IPA) subscale for Autonomy Outdoors) and that their informal caregivers perceived a lower care burden (using a self-rated Burden VAS). In this instance, the program might have been too complex or ambitious, as is demonstrated by the fact that it could not be rigorously applied (e.g., part of the patients and informal caregivers did not receive all key elements of the program, the objectives were not always formulated as per recommendation, the percentage of therapy sessions performed in the patients' home environment was lower than planned, and only about a quarter of the patients and informal caregivers attended the education session).

Nevertheless, there is a need for homogenization of post-stroke rehabilitative care for frail older adults, which will necessarily have to take patient- and caregiver-specific factors into account to ensure feasibility.

## 8. Improvement Versus Quality of Life

The majority of the available evidence studies functional or neurological impairment and recovery, and few data about the quality of life after stroke interventions. The study by Buijck et al. from 2014 [18] demonstrated that high quality of life after stroke rehabilitation, measured with the RAND 36 Health Survey (RAND-36), was associated with high functional independence (Barthel Index for basic ADL and Frenchay Activities Index for instrumental ADL), less neuropsychiatric symptoms (using the Neuropsychiatric Inventory), and less depressive complaints (using the Geriatric Depression Scale). Informal caregiver burden (measured with the Caregiver Strain Index) was not associated with patients' quality of life, but patients' neuropsychiatric symptoms were a significant determinant of high informal caregiver burden. Patients with severe disabilities can have a good overall quality of life, what is called the "disability paradox" and reflects "finding a proper balance between physical, mental, social, and environmental factors, and that this can also be achieved when important life domains are severely affected".

Therefore, while the stroke care community should strive to improve functional independence and neurological recovery, the improvement of quality of life should also be a priority, especially in frail older adults at risk for institutionalization or a high degree of dependence on the informal caregiver.

## 9. Conclusions

Older age is associated with poorer outcomes after stroke, but not with the success of neurorehabilitation. There is no justification for restricting access to neurorehabilitation solely on the basis of age: a thorough evaluation of prestroke independence, neurological status, especially cognitive function, baseline functional impairment, as well as social support, remains necessary to make informed decisions as to what patients will benefit from rehabilitation.

There is a great heterogeneity of settings, methods, study questions, and instruments, and poor replicability and generalizability of results. Ultimately, researchers should address this by studying larger, multi-center cohorts and minimizing referral bias by making sure all or most patients presenting with stroke are screened for inclusion, using validated scales, and adjusting for the different associated factors by using regression analysis.

There is growing evidence that post-stroke outcomes can be predicted by different measures, even very early after the event. A study by Buch et al. from 2016 [19] studied motor improvement in the relationship with corticospinal tract asymmetry using DTI imaging and the Fugl Meyer Assessment 2 weeks after stroke and found that a combination of those two measures predicts impairment improvements after stroke. This was shown for other types of neurological impairments, for example, aphasia, in a study by Nicolo et al. in 2015 [20], that showed that EEG-connectivity of the Broca area with the ipsilesional hemisphere and the contralesional Broca area was correlated with better language improvement.

There is a practical use for this approach using a combination of clinical and physiological data to predict recovery, as demonstrated by Stinear et al. in their 2017 study [21], where the authors developed an algorithm that predicted recovery (poor, limited, good, excellent) using a combination of the SAFE score (shoulder abduction, finger extension) for the upper extremity, age (>80 or <80 years old), the NIHSS Score, and the integrity of corticospinal tracts using motor evoked potential (MEP).

In future studies, both clinical and physiological factors should be taken into account when making predictions to inform the potential for recovery.

**Funding:** This research received no external funding.

**Conflicts of Interest:** The authors declare no conflict of interest.

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
