# Peer review of "Influence of Age on the Success of Neurorehabilitation"

_ctn, doi:10.3390/ctn7010009_

Round 1

Reviewer 1 Report

This review evaluates the influence of age on the success of neurorehabilitation. Although the authors present a clear cut manuscript that is nicely written I miss the following points: 

- Methods: At the moment I dont clearly understand the rationale of this review in the context of the existing knwledge. Please include two/three sentences. This review is a mosaick of the special issue "Neueorehbailitation" I think clear cut eligibility criteria would help the reader to understand how the studies were grouped for synthesis - and at the end to clearly understand the rationale/the choices of the presented topics in the chapters. 

- No limitations i) of the vidence included  nor of the review processed used

- No search strategy?

- Therapy intensity: I miss the factor "Load" here - too often (therapy) "intensity" is only defined/measured with a certain amount of time spent in an active session. However one misses the actual loading of the session and the exertion felt by the participants. Normally the therapy/training intensity/loading of the session can be calculated by the formula: time of session x RPE.  This point has to be included into the discussion.           

Author Response

Author's Reply to the Review Report (Reviewer 1)

First of all, thank you for your insightful comments. The answers to your questions and remarks can be found hereafter.

This review evaluates the influence of age on the success of neurorehabilitation. Although the authors present a clear cut manuscript that is nicely written I miss the following points:

- Methods: At the moment I dont clearly understand the rationale of this review in the context of the existing knwledge. Please include two/three sentences.

Reformulated and elaborated in introduction : “There are stereotypical views as to which older patients benefit less from intensive Neurorehabilitation and increasing pressures from health systems and insurance companies to rationalize care to favor only the most efficient health interventions.”

This review is a mosaick of the special issue "Neueorehbailitation" I think clear cut eligibility criteria would help the reader to understand how the studies were grouped for synthesis - and at the end to clearly understand the rationale/the choices of the presented topics in the chapters.

- No limitations i) of the vidence included  nor of the review processed used

- No search strategy?

This review is a narrative, non-systematic review: the search strategy is not presented, and the studies not grouped according to inclusion/exclusion criteria. Nevertheless, the strategy used was to conduct a literature research using the terms “(Stroke) AND (Neurological Rehabilitation) AND (Outcome Assessment, Health Care) AND (Aged, 80 and over)” in PubMed and then screening abstracts for relevance. The references of the two largest existing reviews from Jongbloed[1] et al. and Black-Schaffner[2] et al were also screened for additional relevant studies.

I added this explanation and specified the type of review article.

- Therapy intensity: I miss the factor "Load" here - too often (therapy) "intensity" is only defined/measured with a certain amount of time spent in an active session. However one misses the actual loading of the session and the exertion felt by the participants. Normally the therapy/training intensity/loading of the session can be calculated by the formula: time of session x RPE.  This point has to be included into the discussion.       

This is an extremely valid point, and only rarely reported/studied. Few of the included studies specified “therapy intensity” in terms of “time spent in an active session” as an explanatory variable. None of the included studies mentioned “load” as a measure of “Intensity (hours) X Rate of Perceived Exertion”, and few studied do, in general. The PubMed search “(Stroke) AND (Neurological Rehabilitation) AND ("Rate of perceived exertion" OR RPE OR exertion)” renders 102 results and the most widely used measure of exertion is the Borg Rating of Perceived Exertion Scale, but there are also physiological measures like percentage of heart rate reserve or even more advanced physiological measures, as exemplified by a recent review from Mah et al on High Intensity Exercise on Lower Limb function[3].  Mostly, this approach equates exertion to cardiovascular effort and not necessarily to “difficulty” of a certain task or therapy, which is a multidimensional concept.

It would be interesting to know if this factor of load or “therapy difficulty” varies with increasing age and it can be postulated that the capacity for effort is more limited in the elderly, which might limit participation. 

I added this point in the chapter “The importance of therapy intensity”.

Please also see the changes made to the original manuscript.

Kind regards.

NB

References:

  1. Jongbloed, L. Prediction of Function after Stroke: A Critical Review. Stroke 1986, 17, 765–776, doi:10.1161/01.STR.17.4.765.
  2. Black-Schaffer, R.M.; Winston, C. Age and Functional Outcome after Stroke. Top Stroke Rehabil 2004, 11, 23–32, doi:10.1310/DNJU-9VUH-BXU2-DJYU.
  3. Mah, S.M.; Goodwill, A.M.; Seow, H.C.; Teo, W.P. Evidence of High-Intensity Exercise on Lower Limb Functional Outcomes and Safety in Acute and Subacute Stroke Population: A Systematic Review. Int J Environ Res Public Health 2022, 20, doi:10.3390/IJERPH20010153.

Reviewer 2 Report

Congratulations, the paper highlights an underreported but highly actual subject in good and comprehensive way. 

The literature includes as well older papers. Strengths and fails are well discussed. However, some recent dutch studies could be included (see attachement). Lastly, comparison with other rehabilitation fields could be interesting.

I would suggest  to discuss a little more the determinant of "improvement". Is it different on older than in younger? You showed very will that improvement is linked with FIM ad admission, but how about the influence of the pre-morbid FIM?

For the future research I suggest to elaborate more facettes  that are missing in practise. E.g. : what are early predictors for good evolution (like the safe model) ? Which measurements should be used ? Is there a need for a international consens (e.g. the stroke recovery and rehabiliation roundtable)?Improvement equal to quality of life?

att:

Vluggen et al. BMC Geriatrics (2021) 21:134 https://doi.org/10.1186/s12877-021-02082-4

Aging & Mental Health, 2014 Vol. 18, No. 8, 980–985, http://dx.doi.org/10.1080/13607863.2014.899969

Author Response

-----------------------

Author's Reply to the Review Report (Reviewer 2)

First of all, thank you for your insightful comments. The answers to your questions and remarks can be found hereafter.

Congratulations, the paper highlights an underreported but highly actual subject in good and comprehensive way.

The literature includes as well older papers. Strengths and fails are well discussed. However, some recent dutch studies could be included (see attachement).

Thank you for the interesting suggestions. I discussed the study by Vluggen et al[1]. in the paragraph “Health disparities in stroke care for older adults” and added a paragraph about quality of life measures where I discussed the study by Buijck et al[2] in the conclusion.

Lastly, comparison with other rehabilitation fields could be interesting.

I would suggest to discuss a little more the determinant of "improvement". Is it different on older than in younger?

The determinants predicting improvement in terms of discharge FIM or rehabilitation efficacy in the different age groups are described in the study by Mutai et al[3]. for example.

There are shared and different factors. For discharge FIM, while there were age-group specific factors (i.e. trunk dysfunction and neglect for the 65-74 age group), premorbid dependence, motor paralysis, and cognitive FIM were shared by the 3 groups. Rehabilitation efficacy depended on recurrent strokes and motor paralysis in the younger group, and on cognitive FIM and non-paretic limb function in the older age-group.

I highlighted this point in the manuscript.

You showed very will that improvement is linked with FIM ad admission, but how about the influence of the pre-morbid FIM?

The only two study that assessed premorbid ADL function were the Study by Colantonio et al. from 1996[4] (premorbid Katz and Rosow scale, premorbid cognitive impairment) and the study by Mutai et al. from 2018[3] (prestroke modified Rankin scale). None of the included studies assessed prestroke FIM.

In the study by Mutai et al. prestroke dependence (mRS 3-5) was associated with lower FIM at discharge from the hospital in all age groups and with rehabilitation efficacy, but only in the intermediate age group. The study by Colantonio showed that ADL function at 6 month post-stroke was associated with prestroke ADL functional impairment in the uni- and multivariate analysis. The risk of institutionalization after discharge was associated with prestroke cognitive impairment and impairment in the Rosow scale in uni- and multivariate analysis, but not prestroke ADL function using the Katz scale.

I included a paragraph on the matter in the manuscript.

For the future research I suggest to elaborate more facettes  that are missing in practise. E.g. : what are early predictors for good evolution (like the safe model) ? Which measurements should be used ?

There is growing evidence that post-stroke outcomes can be predicted by different measures, even very early after the event. A study by Buch et al. from 2016[5] studied the motor improvement in relationship with corticospinal tract asymmetry using DTI imaging and the Fugl Meyer Assessment 2 weeks after stroke and found that a combination of those two measures predicts impairment improvements after stroke. This was shown for other types of neurological impairments, for example aphasia, in a study by Nicolo et al. in 2015[6], that showed that EEG-connectivity of the Broca area with the ipsilesional hemisphere and the contralesional Broca area was correlated with better language improvement.

There is a practical use for this approach using a combination of clinical and physiological data to predict recovery, as demonstrated by Stinear et al. in their 2017 study[7], where they developed an algorithm that predicted recovery (Poor, Limited, Good, Excellent) using a combination of the SAFE score (Shoulder Abduction, Finger Extension) for the upper extremity, Age (> or < 80 years old) the NIHSS Score and the integrity of corticospinal tracts using motor evoked potential (MEP).

In future studies, clinical and physiological factors should be taken into account when devising predictive scores and algorithm to inform the potential for recovery.

I included a paragraph in the conclusion of the manuscript.

Is there a need for a international consens (e.g. the stroke recovery and rehabiliation roundtable)?

As I mentioned, there is great heterogeneity of rehabilitative care for older adults, especially when considering the relatively homogeneous acute stroke care since the implementation of stroke centers. The Study by Vluggen et al[1]. assessed the effect of a multidisciplinary geriatric rehabilitation programme for older persons with stroke on the primary outcome measure of daily activity using the Frenchay Activity Index (FAI) scale and found no significant improvement, but found that patients participating in the programme had a higher level of perceived autonomy of outdoor activities (using the Impact on Participation and Activity (IPA) subscale for Autonomy outdoors) and their informal caregivers perceived a lower care burden (using a self-rated Burden VAS). In this instance, the programme might have been too complex or ambitious, as is demonstrated by the fact that it could not be rigorously applied (e.g. part of patients and informal caregivers did not receive all key elements of the programme, the objectives were not always formulated as per recommendation, the percentage of therapy sessions performed in the patients’ home environment was lower than planned, and only about a quarter of the patients and informal caregivers attended the education session).

Nevertheless, there is a need for a homogenization of post-stroke rehabilitative care for frail older adults, that will necessarily have to take patient- and caregiver-specific factors into account to ensure feasibility.

I added a paragraph in the chapter 7. Health Disparities in stroke care for older adults.

Improvement equal to quality of life?

The majority of the available evidence studies functional or neurological impairment and recovery, and few data about quality of life after stroke interventions. The study by Buijck et al. from 2014[2] demonstrated that high quality of life after stroke rehabilitation, measured with the RAND 36 Health Survey (RAND-36), was associated with high functional independence (Barthel Index for basic ADL and Frenchay Activities Index for instrumental ADL), less neuropsychiatric symptoms (using the Neuropsychiatric Inventory), and less depressive complaints (using the Geriatric Depression Scale). Informal caregiver burden (measured with the Caregiver Strain Index) was not associated with patients’ quality of life, but patients’ neuropsychiatric symptoms were a significant determinant of high informal caregiver burden. Patients with severe disabilities can have a good overall quality of life, what is called the “disability paradox” and reflects “finding a proper balance between physical, mental, social, and environmental factors, and that this can also be achieved when important life domains are severely affected”.

Therefore, while the stroke care community should strive to improve functional independence and neurological recovery, the improvement of quality of life should also be a priority, especially in the frail older adults at risk for institutionalization or a high degree of dependence on informal caregiver.

I added a paragraph under the title “8. Improvement versus quality of life”

Please also see the changes made to the original manuscript.

Kind regards.

NB

References:

  1. Vluggen, T.P.M.M.; van Haastregt, J.C.M.; Tan, F.E.; Verbunt, J.A.; van Heugten, C.M.; Schols, J.M.G.A. Effectiveness of an Integrated Multidisciplinary Geriatric Rehabilitation Programme for Older Persons with Stroke: A Multicentre Randomised Controlled Trial. BMC Geriatr 2021, 21, 1–11, doi:10.1186/S12877-021-02082-4/TABLES/5.
  2. Buijck, B.I.; Zuidema, S.U.; Spruit-Van Eijk, M.; Bor, H.; Gerritsen, D.L.; Koopmans, R.T.C.M. Determinants of Geriatric Patients’ Quality of Life after Stroke Rehabilitation. http://dx.doi.org/10.1080/13607863.2014.899969 2014, 18, 980–985, doi:10.1080/13607863.2014.899969.
  3. Mutai, H.; Furukawa, T.; Wakabayashi, A.; Suzuki, A.; Hanihara, T. Functional Outcomes of Inpatient Rehabilitation in Very Elderly Patients with Stroke: Differences across Three Age Groups. Top Stroke Rehabil 2018, 25, 269–275, doi:10.1080/10749357.2018.1437936.
  4. Colantonio, A.; Kasl, S. v.; Ostfeld, A.M.; Berkman, L.F. Prestroke Physical Function Predicts Stroke Outcomes in the Elderly. Arch Phys Med Rehabil 1996, 77, 562–566, doi:10.1016/S0003-9993(96)90295-6.
  5. Buch, E.R.; Rizk, S.; Nicolo, P.; Cohen, L.G.; Schnider, A.; Guggisberg, A.G. Predicting Motor Improvement after Stroke with Clinical Assessment and Diffusion Tensor Imaging. Neurology 2016, 86, 1924–1925, doi:10.1212/WNL.0000000000002675.
  6. Nicolo, P.; Rizk, S.; Magnin, C.; Pietro, M. di; Schnider, A.; Guggisberg, A.G. Coherent Neural Oscillations Predict Future Motor and Language Improvement after Stroke. Brain 2015, 138, 3048–3060, doi:10.1093/BRAIN/AWV200.
  7. Stinear, C.M.; Byblow, W.D.; Ackerley, S.J.; Smith, M.C.; Borges, V.M.; Barber, P.A. PREP2: A Biomarker‐based Algorithm for Predicting Upper Limb Function after Stroke. Ann Clin Transl Neurol 2017, 4, 811, doi:10.1002/ACN3.488.